# Is the Hitchcock Story Really True? Public Opinion on Hooded Crows in Cities as Input to Management

**DOI:** 10.3390/ani12091207

**Published:** 2022-05-07

**Authors:** László Kövér, Petra Paládi, Isma Benmazouz, Andrej Šorgo, Natalija Špur, Lajos Juhász, Péter Czine, Péter Balogh, Szabolcs Lengyel

**Affiliations:** 1Department of Nature Conservation, Zoology and Game Management, University of Debrecen, Boszormenyi Str. 138, 4032 Debrecen, Hungary; paladi.petra@agr.unideb.hu (P.P.); benmazouz.isma@agr.unideb.hu (I.B.); juhaszl@agr.unideb.hu (L.J.); 2Doctoral School of Animal Science, University of Debrecen, Boszormenyi Str. 138, 4032 Debrecen, Hungary; 3Faculty of Natural Sciences and Mathematics, University of Maribor, Koroska Cesta 160, 2000 Maribor, Slovenia; andrej.sorgo@um.si (A.Š.); natalija.spur@um.si (N.Š.); 4Faculty of Electrical Engineering and Computer Science, University of Maribor, Koroska Cesta 46, 2000 Maribor, Slovenia; 5Department of Economic Analysis and Statistics, University of Debrecen, Boszormenyi Str. 138, 4032 Debrecen, Hungary; czine.peter@econ.unideb.hu (P.C.); balogh.peter@econ.unideb.hu (P.B.); 6Department of Tisza Research, Institute of Aquatic Ecology, Centre for Ecological Research, Eötvös Loránd Research Network, Bem ter 18/c, 4026 Debrecen, Hungary; lengyel.szabolcs@ecolres.hu

**Keywords:** attitude, control, corvids, human-wildlife conflict, urban wildlife management

## Abstract

**Simple Summary:**

Human-wildlife conflicts are a novel topic in urban environments. The recent increase in hooded crows in cities across Europe has increased the frequency of such conflicts, and in some places, the control of crow populations has become a necessity and a hotly debated issue. We surveyed the attitude of people towards hooded crows using an online questionnaire developed to assess their knowledge of crows and which control method is acceptable to most people in Hungary. Many respondents had experience with hooded crows and agreed that their high numbers can cause problems. Most people expressed their willingness to learn about the crows and their management yet did not wish to get directly involved in management activities, which they believed should be the responsibility of professionals. In addition, most people supported the use of non-invasive or less harmful control methods and opposed more intrusive or destructive ones. These results clearly express the difficulty in identifying the most suitable and tolerable way to manage urban crow populations and thus address emerging human-wildlife conflicts in urban environments.

**Abstract:**

In recent years, the Hooded crow (*Corvus cornix*) has become one of the most successful wild bird species in urban environments across Europe. Hooded crows can cause several problems in cities, including trash scattering, noise disturbance, and aggressive behavior toward humans or pets, and they can be potential vectors of pathogens. To find effective solutions, the public has to be involved in the decision-making process in urban planning management, managed by the city administration. In this study, we surveyed the attitude of people in Hungary towards crows and crow management by collecting information using an online questionnaire containing 65 questions published in 14 Facebook groups. We found that many people were familiar with corvid species and had personal experience with them. In most cases, these experiences were not negative, so the crows were not or only rarely perceived to cause problems to people, such as aggressive behavior, damage to cars or stealing something. Most respondents recognized that the presence of large numbers of hooded crows is a problem to be solved and acknowledged that they do not know how to resolve it. The majority of people expressed their interest in raising public awareness of crows but not in their management actions, which they believe should be implemented by experts. Most respondents preferred passive, harmless methods. More direct methods such as egg/chick removal from the nest, control by trapping, poisoned baits or firearms, or oral contraceptives were the least acceptable. These results express the difficulty in identifying a control method for managing hooded crow populations that is both acceptable to most people and effective at the same time. This study demonstrates the importance of involving public opinion in wildlife management and providing more information to citizens to reduce human-crow conflicts.

## 1. Introduction

As urban areas are growing and expanding globally, human-wildlife conflicts are becoming more frequent and even inevitable in places [1,2]. This challenge has led to the recent emergence of the field of urban wildlife management, focusing on the study and management of animal species living in human-populated areas [3]. One major theme in urban wildlife management is to identify animal species that cause problems to citizens of urban areas by making a lot of noise, leaving fecal droppings that can spread disease, strewing garbage around, causing damage to buildings and landscaping, and attacking humans and pets [4]. This theme involves the study and assessment of the ecological characteristics, mainly the habitat requirement, reproduction and spread of the problem species, the development and implementation of management plans, and the evaluation of citizens’ attitudes and raising their awareness towards the given species, the associated problem and the options to solve or reduce it by the generation and dissemination of relevant knowledge [5,6]. As an important interplay between natural and social sciences, urban wildlife management requires the involvement and cooperation of the affected general public and wildlife professionals for effective actions [4]. Currently, there is a growing interest in community-based conservation approaches as traditional conservation is shifting towards a “people and nature” approach based on a new and deeper understanding of social dimensions. Therefore, engaging with social science in order to better understand human–nature connections has become increasingly practiced.

Numerous bird species (e.g., corvids, [7,8]) have successfully colonized and spread in urban environments, and many are causing a wide range of problems around the world by damaging agricultural crops, parks or other urban infrastructures, causing aircraft accidents, exerting predation on desirable species, and showing hostile behaviour to humans [9]. Accordingly, various methods have been developed to manage bird populations growing too large [10,11]. One straightforward way is to address the primary cause of growth; for example, the feeding of pigeons by citizens can be banned [12]. Scaring birds away is often done by visual alarms such as bright, flickering materials, silhouettes or decoys of predatory birds, acoustic alarms by sounds such as alarm calls, predator calls, sound cannons or ultrasounds, or by electromagnetic fields. Visual and acoustic alarm methods can also be combined, e.g., by using fireworks to prevent bird damage to crops [13,14,15]. Physical alarms by humans, trained dogs or birds of prey are also common. Recently, drones have been increasingly applied to alarm birds [16,17,18]. When alarm methods do not suffice, physical structures to keep birds away from their nests or roosting sites are often used. Direct interventions such as shaking or pricking the eggs or their treatment with various substances can be effective at reducing reproductive success [19,20,21,22]. Trapping and translocating individuals can also reduce the number of unwanted birds [2]. However, this method is costly and may not be effective due to the influx of birds from other areas [23]. More drastic methods such as the sterilization of breeding-age individuals with hormones and other agents, sometimes surgically, have also been applied [24]. Finally, lethal methods, including poisoning by various chemicals [25] or shooting by firearms [26,27], have also been used for several decades. However, in many cases, these population management actions are difficult to implement in urban environments. Some of the methods, especially the lethal ones, are rather ethically unacceptable to citizens and sometimes prohibited in cities [14]. For example, audible alarm calls can be annoying to humans or their pets, sterilizers/poisons may threaten non-target species, and translocation may cause new problems at the site of release. It follows that any planned population management actions need to be made public ahead of implementation, thoroughly discussed with interested citizens, and adjusted as necessary to increase the acceptability of the actions, which requires an understanding of citizens’ attitudes toward the problem species and management options [23].

The Hooded crow (*Corvus cornix* Linnaeus, 1758) is a large crow species widespread in much of Eurasia. This species used to be a typical bird of rural agricultural areas; however in recent decades, this species has colonized large cities, rapidly adapted to urban environments, and increased in number in many cities in Europe. Hooded crows are known to have outstanding cognitive abilities and general intelligence among birds [28], which make them especially successful in adapting to new environments. Cities provide several resources for birds that can exploit them, including a permanent or near-permanent food supply, abundant nesting sites, and a low number or absence of predators and competitors [29,30]. For example, open-air zoos with year-round food sources have likely been the centers of colonization in several European cities [31]. After settling in a new location, the population size often increases rapidly, sometimes exponentially. For instance, the number of nesting pairs increased considerably over the years in the city of Debrecen (Hungary, Europe), co-occurring with a tendency to use an increasingly wider range of nesting sites, which attests to the adaptability of the species [7]. However, citizens are not always tolerant of the presence of the species [32]. Hooded crows are keen predators of eggs, nestlings and sometimes adult individuals of smaller birds [33,34], usually admired by many people. Crows can also be noisy at times, and permanent noise is also a cause of concern [35]. During the breeding season, hooded crows can be aggressive around their nests and will attack humans and their pets by dive-bombing them [23,32,36]. Hooded crows regularly feed on garbage dumps and sewage treatment plants, creating a huge mess and possibly spreading potential pathogens [37]. Finally, their presence at airports is particularly dangerous due to possible collisions with aircraft [38]. In addition, they can cause damage to motor vehicles and various buildings by leaving messy droppings and accumulating nesting materials that can, for example, damage roofs and block ventilation systems [39,40]. With regard to these problems, the management of urban populations of hooded crows will be necessary in the future.

Interventions to manage the urban populations of hooded crows necessarily take place in the city, and it is essential to plan such actions with the involvement of interested citizens or citizen groups. Beyond providing information to the public, it is also necessary to collect information on how the affected citizens relate to potential interventions and their specific methods. An intervention that exceeds the tolerance level of the residents can be detrimental to both the relationship between people and management authorities and the judgment of the institution and personnel implementing the actions [41]. However, people’s opinions regarding wild species and their management can be influenced by a number of factors. Attitudes can vary depending on age, gender, personality type, whether the individual has pets, eats meat, and so on [42,43,44,45,46]. In addition, different species are judged differently as there are more popular and less popular species and species groups [47]. For example, squirrels, which are revered by many people, are major predators of bird eggs and nestlings, yet the general judgement of this species is very different from that of “harmful” crows that “kill songbirds”. The perception of crows among humans is generally bad partly because of such beliefs, which are further exacerbated by various products of popular culture. For instance, in the famous cult movie entitled ‘The Birds’, directed by Alfred Hitchcock in 1963, corvids (ravens), as well as gulls, attacked and killed people.

The aim of this study was to assess the social judgement of hooded crows and their possible management in Hungary, where many cities have been relatively recently colonized by the species. Our specific objectives were to assess (i) the attitude of city-dwelling humans to crows, (ii) how citizens relate to possible crow population management and (iii) what methods are considered acceptable for the management/control of urban crow populations. While our knowledge of the colonization, the increase and nest-site selection of hooded crows in urban environments [7], the movement patterns, the habitat use [48] and the trapping methods to capture crows [49] has been increasing, there is a need to recognize the importance and assess the public opinion about city-dwelling crows and their management in places where such management becomes necessary and/or is already planned.

## 2. Materials and Methods

### 2.1. Data Collection 

We collected data in a questionnaire-based survey conducted between 22 May and 27 June 2019. The questionnaire was designed based on a questionnaire used in a previous study in Slovenia [41], with modifications agreed upon in consultations with social scientists in Hungary (see below). The questionnaire was published in local, settlement-based Facebook groups. These social media groups had already existed (e.g., ’I heard in town’ group), and they were not specific or related to any environmental topics. We selected these from three pools of groups: (i) cities serving as county seats, (ii) other cities with at least 5000 inhabitants, and (iii) cities with less than 5000 inhabitants. The main consideration in the selection was that the group had to display online activity in the months before the study. We then sent requests to post the questionnaire to the administrators of each selected group. Our request was granted, and the questionnaire was posted in a total of 14 Facebook groups, which involved groups from six of the seven most populous cities in Hungary (Figure 1). Each of these cities has seen increased numbers of breeding pairs of hooded crows in the last 10–20 years. For example, the number of nesting pairs increased from around 10–15 pairs in 2006 to 75 pairs in 2012 in Debrecen, the second-largest city [7]. After the posting of the questionnaire, the response was voluntary and anonymous in all cases. To facilitate participation and complete responses, if the respondent provided an email address, the questionnaire survey was associated with a sweepstakes with a wellness weekend for two as the main prize.

### 2.2. The Questionnaire

The questionnaire was designed to include five main sections as in ŠPUR et al. (2016) [41]. The “Demographics” section included questions about the personal details of the respondents: gender, age (four categories: <19, between 19 and 39, between 40 and 59, 60< years), highest level of education (elementary school, high school, higher vocational education or bachelor’s degree, master’s degree or PhD degree), place of residence/workplace or educational institution (capital, county town, other urban, rural), and membership in a hunting association or in nature conservation or animal welfare organizations. 

The “Species knowledge” section included a quiz of six corvid species (magpie *Pica pica*, jay *Garrulus glandarius*, jackdaw *Corvus monedula*, raven *Corvus corax*, hooded crow, and rook *Corvus frugilegus*) with six possible answers. 

The “Negative experiences with the Hooded Crow” section included 15 statements on potential negative experiences, with the following answer options: “I have personal experience”, “I heard from others”, “I heard in the media”, and “I have no experience”. 

The “Coexistence with Hooded Crows” section sought responses to 21 statements about the attitude to hooded crows. Potential responses ranged on a Likert scale from 1 to 5, where 1 was “strongly disagree” and 5 was “strongly agree”. To eliminate automatic responses, 11 statements were negatively worded. 

Finally, the “Attitudes towards population management methods” section listed 16 drastic and less drastic population management methods, and respondents had to mark acceptability on a Likert scale of 1 to 5 to show how much they agreed with a particular method. Answer 1 was “completely unacceptable,” and answer 5 was “completely acceptable”.

### 2.3. The Respondents 

We aimed at the widest possible representation of Hungarian cities, and we obtained responses from the capital city (Budapest), the four largest cities (Debrecen, Szeged, Miskolc, Pécs), the 6th, 12th and 18th largest cities and six smaller cities, which provided a meaningful cross-section of the cities of various sizes in Hungary. However, it should be noted that due to the online nature of the questionnaire, it was accessible only to those who had access to the internet and were active members of the social media interested in the topic. The results thus do not necessarily reflect the views of non-electronic media users and cannot be generalized to the whole population. However, our approach also had benefits because it was more likely that only people who had been genuinely interested in the topic completed the questions. These people will likely be those who will be most interested in crow-related interventions in the future and who may thus influence the course of population management actions. A total of 1752 questionnaires were completed. The questionnaire was incompletely filled in 12 cases; these were excluded from the analysis. Accordingly, we processed a total of 1740 questionnaires. A detailed description of our sample is shown in Appendix A.

### 2.4. Statistical Processing of Data

Collected data received from 1740 respondents were initially checked for missing data and outliers. The data entered in the questionnaires were coded for easier processing. Based on frequencies, measures of central tendencies (mean, standard deviation and median) were calculated. To examine the independence between categorical (nominal or ordinal) variables, we built contingency tables and performed Chi-squared tests [50,51]. The results of the statistical tests were considered significant at *p* < 0.05. To assess the different attitudes of the respondents, we developed dimensions to represent these attitudes using principal component analysis (PCA). First, we calculated Cronbach’s alpha index (as it is the most common measure of internal consistency; if the coefficient of 0.70 or higher, it is considered “acceptable” in most social science research situations) [52], the Kaiser–Meyer–Olkin (KMO) value (values between 0.8 and 1 indicate the sampling is adequate), and the Bartlett test for sphericity. From the significant relationships (*p* < 0.05), the optimal MSA (measure of sampling adequacy), KMO values (>0.5), and the significant Bartlett test (*p* < 0.05), we concluded that the variables were highly correlated and suitable for dimension reduction by PCA. In determining the number of principal components, we first considered the Kaiser criterion, which states that the eigenvalue of the principal components must be at least one [53]. With the PCA, we developed dimensions representing the different attitudes of the participants in the sample [54]. Further analysis of the components was performed by using two-step cluster analysis, which has several advantageous properties compared to traditional methods (e.g., it can be used effectively for the analysis of large databases and provides an efficient cluster number search algorithm for users). In the case of the two-step clustering, we used an automatic cluster number search algorithm suggested by the literature. This suggested the choice of a two-cluster solution, which was also supported by the value of the silhouette measure of cohesion and separation (value above 0.0) [55]. We used IBM SPSS Statistics 25.0 software to analyze data [56].

## 3. Results

### 3.1. Species Knowledge

Of those who completed the questionnaire, 80% correctly identified the magpie, 84% identified the jay, 76% identified the hooded crow, 1135 (65%) identified the jackdaw, 1063 (61%) identified the raven, and 59% identified the rook. A total of 41% of respondents successfully recognized all species. There were few differences in species knowledge between the genders and age groups (Appendix A). 

### 3.2. Negative Experiences with the Hooded Crow

The most commonly observed negative behaviors by hooded crows were the consumption of fruit or walnuts (46%), giving out frightening calls (44%), and rummaging in trash (37%) (Table 1). More serious incidents, such as damaging field crops or garden products, stealing something, eating eggs/nestlings or small birds, attacking domestic animals or adult persons, and damaging cars or buildings, were mostly heard from other people by respondents (Table 1).

### 3.3. Coexistence with Hooded Crows

A high proportion (>50%) of respondents expressed agreement with three statements implying crow population control, and the proportion of agreement was between 36% and 49% for two other statements implying population control, i.e., a sign of support for crow control was found for 5 of 21 statements. More than half of the respondents agreed with the statements (19, 1, 20) that the “Crow population control is beyond my scope and should be the business of experts” (69% when percentages for “Strongly agree” and “Agree” responses were combined), “I would like to be involved in projects that aim to raise attention and awareness to crows” (57%), and “Urban crow numbers should increase as they contribute to the diversity of the urban areas” (59% when percentages for “Strongly disagree” and “Disagree” responses were combined). In addition, a proportion of the respondents favoured crow control by agreeing (statements 2 and 15) that “Colonization of cities by crows is a problem that should be solved” (44% when percentages for “Strongly agree” and “Agree” responses were combined), and “City crows should enjoy unlimited legal protection” (39% when percentages for “Strongly disagree and “Disagree” responses were combined) (Table 2).

In favour of crow protection, more than half of the respondents disagreed with the control and removal actions of hooded crows in urban areas. Furthermore, 36–50% of responders accepted the experts’ management efforts, while some believe that crows should be protected as any wild bird species (38–47%). These results suggest that the respondents were generally opposed to the idea of crow population control. 

Finally, there were statements (11, 21) for which the proportion of agreement and disagreement was quite similar (“I would sign a petition against measures to control the population of crows” (35% agreed, 37% disagreed), “I am indifferent to crows; I have no interest in them or any problem with them” (33% agreed, 38% disagreed)). For the remaining three statements (14, 16, 18), the proportions of agreement or disagreement were both below 33% (Table 2).

The mean score of agreement differed between men and women for 14 of the 21 statements (Appendix A). For statements implying crow protection, women expressed higher scores of agreement than men in seven of nine cases, and for statements in favour of crow control, women expressed higher scores than men in four cases (Appendix A).

The same was examined among hunting company members and non-members, and we found significant differences for 18 of the 21 statements (e.g., “Crows should be de-listed as game species, which allows unlimited control measures”.) (Table 3). For statements of crow protection, we found higher scores for non-members in every case. For near-neutral statements (Statements 7, 21) (e.g., “I am indifferent to crows; I have no interest in them or any problem with them”.), agreement scores by hunting company members were higher than those by non-members. Finally, among statements for crow control, we found that hunting company members expressed higher scores in three cases (14, 18, 19) (e.g., “Crows should be de-listed as game species, which allows unlimited control measures”.) (Table 3). 

In the PCA, three components could be distinguished (Figure 2). The first principal component (PC) explained 26.19% of the variance (Cronbach’s α = 0.71, eigenvalue = 4.98), and could be related to (dis)interest in crows. The second PC explained 20.48% of the variance (Cronbach’s α = 0.81, eigenvalue = 3.89), and could be linked to the population management of crows. The third PC explained 8.54% of the variance (Cronbach’s α = −0.78, eigenvalue = 1.62) and represented openness to different crow projects.

### 3.4. Attitudes towards Population Management Methods

Only one method of population management was generally acceptable for the respondents, which was “Scaring crows using techniques that do not cause noise” (sum of “Acceptable” and “Completely acceptable” percentages: 50.5%) (Table 4). This was followed by other scaring techniques (“All measures for scaring crows”, 28%; “Scaring crows with noise”, 27%), other measures (“All measures to control the number of adult crows by authorized persons”, 27%; “All measures to control nesting success”, 26%) and the use of traps (in urban areas, 22%; on farmland or in the countryside, 19%). We note that the last six methods also had high values for unacceptability (range of the sum of “Completely unacceptable” and “Unacceptable” values: 40% to 61%). All the other methods had much higher values for unacceptability (range: 62% to 92%), with “Setting poisoned baits for crows” (91%) and “Shooting crow chicks in the nest” (92%) being the most opposed methods.

In the PCA, two components could be distinguished (Figure 3). The first PC explained 26.46% of the variance (Cronbach’s α = 0.85, eigenvalue = 4.23), and the factors were related to more drastic (hard persecution) population management methods. The second PC explained 24.74% of the variance (Cronbach’s α = 0.84, eigenvalue = 3.96), and mostly incorporated less drastic (soft persecution), alarm-based methods.

### 3.5. Clustering Analysis Based on the Results of the Principal Component Analysis

In order to gain deeper information about public opinion on hooded crows, we used a two-step clustering method by using the components that we determined based on coexistence (OP) and attitude (ME) statements. The mean values of the clusters are shown in Table 5.

The results in Table 5 show that there was a significant difference between the clusters in the assessment of all five dimensions. While respondents in the first cluster showed stronger agreement on “coexistence” dimensions, they did not agree on statements neither about hard nor soft persecution population control approaches. In the case of the respondents of the second cluster, respondents tended to disagree with the statements of “coexistence” and showed a strong agreement on statements related to both hard and soft persecution approaches.

In the next step, we wanted to determine if there was a significant relationship between clusters and species knowledge. The results of the contingency table analysis are shown in Table 6.

We found a significant relationship between species knowledge and the clusters ($\chi^{2\;}$= 9.74; df = 1; *p* = 0.002) because we found significantly fewer respondents in the second cluster who recognized all species. From this, we can conclude that respondents with weaker species knowledge were more reluctant to coexist with crows, and in addition, they were more supportive of the control of their populations.

## 4. Discussion

### 4.1. Evaluation of Results 

Our study provides several results that are relevant to understanding urban people’s attitudes towards hooded crows and corvids in general. Species knowledge was 60% or more for all corvid species. The highest values, 84% and 76%, were found for the two species, magpie and hooded crows, respectively, that are best adapted to urban environments in Hungary. For our focus species, the hooded crow, older age groups and men were slightly better at recognizing the species than younger age groups and women. Direct negative experiences were reported less frequently than could be expected, and negative experiences reported represented relatively low disturbances (eating fruit, walnut, trash, frightening calls) compared to other, more serious incidents (attacking persons, stealing food from people, damaging things etc.), which were usually heard by the respondents from other people. The most important known biases of this study arise from the distribution of the respondents by age and gender as more females than men and a higher number of younger than older people filled out the questionnaire.

A total of 33% of respondents said they were interested in hooded crows at a certain level. We found serious extremes in people’s attitude, as there were those who especially liked crows, while others were very passionate about them. However, based on the results, it can be stated that the attitude of the respondents towards crows is mostly not determined by their own experience but by second-hand information. Only a fraction of respondents had their own experience with a more serious incident. An awareness-raising campaign on the subject may make sense, as second-hand information may not be accurate in all cases, and this may lead to a misconception about hooded crows. Dissemination of knowledge can also be important because of the opposites in the results. Such a campaign itself may even be feasible, as the majority of respondents said they would like to take part in projects aimed at raising public awareness and raising awareness about crows. Based on the results, respondents are aware of the problems caused by crows but do not know how to solve them. Due to being aware of all this, it is likely that the majority of the inhabitants would accept efforts to manage the population of hooded crows, but caution should be exercised in choosing the method and informing the people. The majority of respondents would not participate in population management projects as they do not feel it is their own area of competence; they would rather entrust the implementation to experts. However, the choice of method would not be left to the expert, so the question is: if there is a real need to control the population of crows, what method can we use that is both effective, applicable in an urban environment, and acceptable to the public? Based on the results so far, it is difficult to answer this question. Of the methods, the most accepted by the public (50.5%) would be some kind of alarm method that is not noisy. However, this would not necessarily be effective for hooded crows, who, due to their high level of intelligence, quickly learn what real danger is and is not for them. One such possible method could be the use of drones [57]. This was followed by alarm methods and then by some way of controlling nesting success (26.3%). This is again contradicted to some extent by the fact that the removal of eggs and chicks from nests was less accepted among respondents (7.5%). Unsurprisingly, the most drastic methods became the least accepted: placement of poisoned baits (91.2% opposed) and control by gun both in the wild and in the city (64.7% and 83.0% disagreed). By implication, these could not even be applicable methods for population management in urban environments. According to the Hungarian Game Management Law (1996/LV), the administrative interior of settlements does not qualify as a hunting area, so control by gun is not possible there either. The placement of poisoned baits is also an illegal form of control, according to the Bern Convention. However, the sterilization of crows is used in several countries [58], so there would be a good chance that it would be feasible in Hungary as well; however, 67.9% of the respondents did not agree with it. In light of all this, it would be difficult to select and implement a method that is both effective and accepted by the people. From the point of view of citizens, perhaps the best solution could be the use of traps (e.g., ladder-trap) or a drone alarm, for they do not cause harm to birds, but they can be effective for alerting and controlling the population. 

### 4.2. Comparison of Results

We compared our results to those of a previous study carried out in Slovenia [41], where the attitudes of the city inhabitants towards Hooded Crows and different population management methods were examined. The proportion of respondents was similarly distributed in Slovenia, both in terms of gender and place of residence/workplace and the highest level of education. There were pre-existing differences in species knowledge because a higher proportion of respondents recognized the Hooded Crow in Slovenia (95%). For the other species, the Slovenian respondents also performed better. There was a significant difference according gender in both studies. There were also differences on issues related to negative experiences. After examining the chapter of the questionnaire entitled Coexistence with Hooded Crows, we discovered an even more conspicuous difference between the respondents’ interests in the topic in the two countries. However, overall, both studies were successfully completed, respondents were interested in the topic, and many of them would also be active participants in the crow awareness project. Unfortunately, none of the studies brought results that would provide clear guidance in dealing with the problems associated with hooded crows. Based on the responses to the questionnaire, there is no specific method that is effective for managing the population of hooded crows and that the people also consider fully acceptable. Aldo Leopold has recognized the need to include the general public in wildlife management decision-making since the 1940s. [59]. Additionally, though human dimension studies have come a long way, still, today managers are facing challenges in implementing such aspects in their decision-making processes. Considering the importance of people’s perspectives, the acquisition of sound data that displays such prospective in regards to wildlife and wildlife management is necessary; however, making use of such information creates challenges for managers, since, in the end, the use of this information in decision making mainly lies on the quality of such data [60]. Additionally, it is very important to understand the reasons behind people’s different attitudes towards wildlife and wildlife management approaches. Accordingly, it is agreeable to say that for successful wildlife management, regardless of the habitat, public involvement should be proactive, representative of the entire consistency and fully integrated into decision making from the conception to the implementation of such projects.

Although most studies involving public opinion in decision-making have been theoretical until now, few wildlife researchers have had the opportunity to implement and thus evaluate the role and potential success of different stakeholders in wildlife management projects. Stout et al. (1996) [61] have examined different ways of directly or indirectly involving different groups of people, ranging from farmers and conservationists to the general public, in the white-tailed deer management plan. The authors expressed the importance of public involvement, especially during the participation process itself, as they believe the most adequate way is what they called “the citizen task force approach,” in which the general public is an essential part of regular meetings where information is shared between the different parties and where education and increased awareness is a key feature. In Norway, Tombre et al. (2013) [62] concluded that only adequate negotiation between the different stakeholders could allow them to arrive at an agreement and potential successful management plan for increasing goose populations. Yet, after long years of surveys and information gathering, wildlife managers are still challenged.

### 4.3. Suggestions

An important aim of the study, in addition to examining the general attitude of citizens towards urban hooded crows, was to assess which population management methods can be considered acceptable among the respondents. Based on the results, the people would prefer passive methods that do not mean the death of birds, and many would prefer to influence the efficiency of nesting rather than kill already hatched or adult individuals.

In our opinion, the most important step should be to inform the inhabitants. Since the emotionally charged attitudes towards wildlife differ according to species and context, having better knowledge of the species in question may result in more appropriate actions. It would be important to familiarize them with the lifestyle and behavior of crows living in the city, specifically how to approach them during the more critical nesting period so as to avoid possible attacks. It would also be particularly important to inform the people about the role of free access to waste (as a source of food), which can greatly help crows to survive. If waste management and its placement were more strictly regulated, the number of human-crow conflicts could be reduced (without population management) in the long term [23], which could be a more acceptable option for residents who oppose population management.

Taking into account the opinions of the respondents, we would suggest two possible population management methods. One such method is to be more “hidden” from the eyes of the citizens through the use of traps in places frequented by hooded crows (e.g., closed parts of parks, zoos, botanical gardens). This can be a method of population management that is legally feasible within the city as well; the use of traps is selective and does not result in the immediate death of trapped birds. Previous research has shown that it can be particularly effective in reducing the number of crows in an urban environment, e.g., application of the ladder trap [49]. 

Another possible method is to alarm crows by using drones [57]. This is fully in line with the views of respondents, as most would prefer an alarm method that does not involve a lot of noise. Drones are increasingly being used to alarm various problem bird species (e.g., gulls, pigeons) [17,18,63]. One of its advantages over other alarm methods is that it is a mobile, automatic device. As a result, it could be targeted where the presence of crows is currently the biggest problem. In several cases, the alarming effects of drones have been enhanced by designing the structure in a predator-like mode [64,65]. In our opinion, the use of drones can be an effective method for population management in the case of areas and buildings where the presence of crows is not desirable; all of these are acceptable by the citizens. It is obvious that public involvement should be mandated by wildlife managers; thus, managing parties should develop a coherent communication plan, clearly outlining the objectives, goals, different approaches and time frames.

### 4.4. Conclusions

Our survey of public opinion on hooded crows and the people’s attitude toward crow population management approaches provides a better understanding of the human perception of corvids in cities and important insight for the effective management of crow populations. Survey respondents reported direct or, more often, indirect experiences with various problems that hooded crows caused, confirming that people perceived crows as a nuisance. Respondents supported non-invasive or less harmful control methods and were less tolerant of more intrusive or destructive methods. This study provides a better understanding of human perceptions of human-wildlife conflicts in general and human-corvid conflicts specifically and points to management options that are both acceptable to the public and can help to reduce the impacts of human-wildlife conflicts in urban environments. A more aware and better-educated public opinion can greatly facilitate management actions as well as provide important insights for such inclusive management.

## Figures and Tables

**Figure 1 animals-12-01207-f001:**
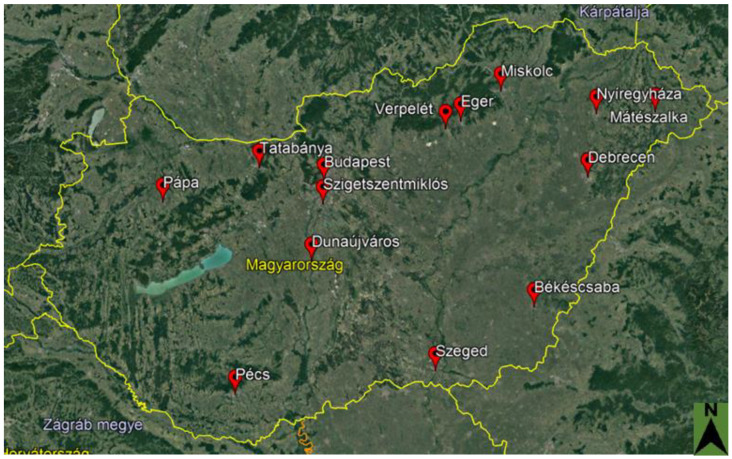
Locations of local community Facebook user groups in which the questionnaire was published. Source: Google Earth.

**Figure 2 animals-12-01207-f002:**
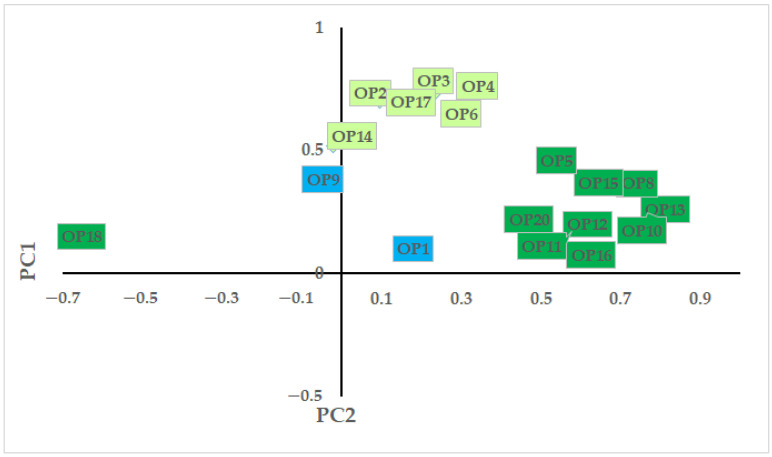
Results of the principal component analysis on responses to statements on coexistence with hooded crows. Note: total explained variance: 55.2%; Bartlett’s test of sphericity: *χ*^2^ = 14219.47, *p* < 0.001; KMO = 0.932; Cronbach’s α = 0.815. OP1: I would like to be involved in projects that aim to raise attention to and awareness of crows. OP2: Colonization of cities by crows is a problem that should be solved. OP3: For the management of crow numbers, all measures by experts are acceptable. OP4: I find measures to control crow populations acceptable and support them. OP5: Damages caused by crows are minor and do not justify population control measures. OP6: The number of crows should be reduced regardless of the type of their habitat. OP7: Crows should not be bothered as their numbers will reach a natural balance. OP8: We should protect crows regardless of the type of their habitat. OP9: I would like to participate in projects aiming to control the population of crows. OP10: The hooded crow is just one of many bird species that should enjoy unlimited protection. OP11: I would sign a petition against measures to control the population of crows. OP12: Damages caused by crows should be reimbursed but should not justify population control. OP13: Crows should be de-listed as game species, which would make their protection easier. OP14: Only the numbers of crows living in cities should be reduced. OP15: City crows should enjoy unlimited legal protection. OP16: Claims to control populations come from the hunting lobby, who aim to shoot more crows. OP17: Crows are wild and do not belong in cities, so they should be removed from urban areas. OP18: Crows should be de-listed as game species, which allows unlimited control measures. OP20: Urban crow numbers should increase as they contribute to the diversity of the urban areas.

**Figure 3 animals-12-01207-f003:**
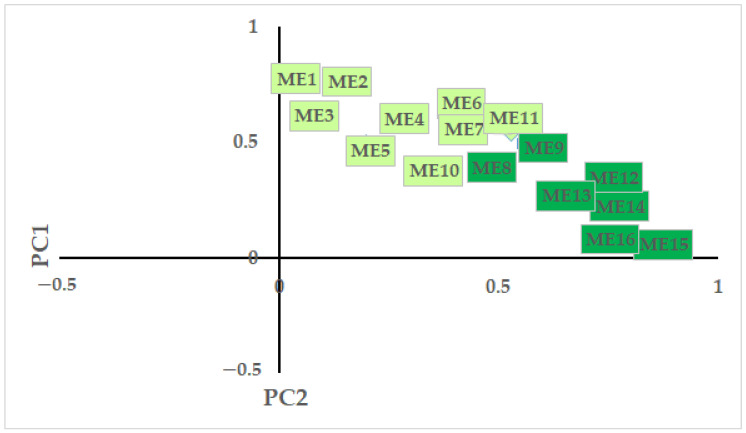
Results of the principal component analysis on responses regarding the acceptability of management methods of hooded crow populations. Note: total explained variance: 51.2%; Bartlett’s test of sphericity: *χ*^2^ = 11703.235, *p* < 0.001; KMO = 0.918; Cronbach α = 0.897. ME1: Scaring crows using techniques that do not cause noise. ME2: All measures for scaring crows. ME3: Scaring crows with noise. ME4: All measures to control the number of adult crows by authorized persons. ME5: All measures to control nesting success. ME6: Use of traps to capture crows in urban areas. ME7: Use of traps on farmland or in the countryside. ME8: Persecution of breeding crows using domesticated birds of prey (falconry). ME9: Physical removal of crows’ nests in urban areas. ME10: Chemical sterilisation of crows. ME11: Shooting adult crows on farmland or in the countryside. ME12: Removal of chicks or eggs from nests. ME13: Shooting adult crows in urban areas. ME14: Crows should be considered as rats, and could be eliminated by anyone. ME15: Shooting crow chicks in nests. ME16: Setting poisoned bait for crows.

**Table 1 animals-12-01207-t001:** Percentage of responses reporting negative experience with hooded crows (listed in decreasing frequency of personal experience).

Negative Experience	Personal Experience	Heard from Others	Heard in Media	No Experience
Eating fruit or walnuts	46.4	12.1	4.7	36.8
Giving frightening call	44.4	10.1	3.7	41.8
Ripping and rummaging through garbage bags	37.0	13.4	6.7	42.9
Damaging garden products	17.8	22.5	6.1	53.6
Contaminating goods with faeces	15.8	11.7	4.5	68.0
Eating a small bird	14.0	18.0	5.6	62.4
Damaging field crops	13.5	24.2	10.3	52.0
Eating eggs or chicks in a nest	11.7	19.0	8.0	61.3
Stealing something	10.3	19.5	11.2	59.0
Attacking an adult domestic animal (dog, cat etc.)	9.9	14.5	10.1	65.5
Attacking an adult person	5.0	12.3	10.8	71.9
Damaging residential buildings	4.4	10.3	5.3	80.0
Damaging a car	3.3	10.7	4.0	82.0
Killing and eating a young domestic animal	2.5	10.4	4.0	83.1
Attacking a child	2.4	9.4	8.2	80.0

**Table 2 animals-12-01207-t002:** Percentages, mean, median and S.D. of responses to statements on coexistence with hooded crows. Statements highlighted in grey are negatively coded and interpreted. * In cases of highlighted statements, reverse values (mean and median) are presented.

Statement	Strongly Disagree	Disagree	Neutral	Agree	Strongly Agree	Mean Score *	Median Score *	S.D. Score
1. I would like to be involved in projects that aim to raise attention for and awareness of crows.								
	8.3	8.4	26.3	22.0	35.0	3.67	4.0	1.260
2. Colonization of cities by crows is a problem that should be solved.								
	13.3	12.5	30.2	19.4	24.6	2.71	3.0	1.323
3. For management of crow numbers, all measures by experts are acceptable.								
	30.2	19.3	23.7	13.2	13.6	3.39	3.0	1.386
4. I find measures to control crow populations acceptable and support them.								
	18.0	18.4	35.1	14.3	14.2	3.12	3.0	1.266
5. Damages caused by crows are minor and do not justify population control measures.								
	10.2	14.0	36.4	19.0	20.4	3.26	3.0	1.220
6. The number of crows should be reduced regardless of the type of their habitat.								
	45.3	21.0	23.4	5.8	4.5	3.97	4.0	1.152
7. Crows should not be bothered as their numbers will reach a natural balance.								
	9.3	12.2	29.1	22.6	26.8	2.55	3.0	1.260
8. We should protect crows regardless of the type of their habitat.								
	9.0	13.9	34.8	19.7	22.6	3.33	3.0	1.221
9. I would like to participate in projects aiming to control the population of crows.								
	40.1	17.7	25.2	8.6	8.4	3.73	4.0	1.295
10. The hooded crow is just one of many bird species that should enjoy unlimited protection.								
	13.2	13.9	34.7	19.1	19.1	3.17	3.0	1.264
11. I would sign a petition against measures to control the population of crows.								
	22.6	14.0	28.3	15.8	19.3	2.95	3.0	1.404
12. Damages caused by crows should be reimbursed but should not justify population control.								
	12.5	11.6	31.3	23.2	21.4	3.30	3.0	1.272
13. Crows should be de-listed as game species, which would make their protection easier.								
	11.8	10.3	30.9	17.9	29.1	3.42	3.0	1.319
14. Only the numbers of crows living in cities should be reduced.								
	19.0	13.1	36.3	21.7	9.9	3.10	3.0	1.224
15. City crows should enjoy unlimited legal protection.								
	18.6	20.3	35.1	12.4	13.6	2.82	3.0	1.258
16. Claims to control populations come from the hunting lobby, who aim to shoot more crows.								
	19.3	11.9	38.5	14.7	15.6	2.95	3.0	1.289
17. Crows are wild and do not belong in cities, so they should be removed from urban areas.								
	39.5	20.9	25.1	9.8	4.7	3.81	4.0	1.194
18. Crows should be de-listed as game species, which allows unlimited control measures.								
	17.4	13.8	42.5	12.3	14.0	3.08	3.0	1.228
19. Crow population control is beyond my scope and should be the business of experts.								
	4.7	4.4	22.2	18.4	50.3	1.95	1.0	1.148
20. Urban crow numbers should increase as they contribute to the diversity of the urban areas.								
	35.7	22.8	32.9	5.1	3.5	2.18	2.0	1.083
21. I am indifferent to crows; I have no interest in them or any problem with them.								
	21.7	16.0	29.3	16.8	16.2	3.09	3.0	1.353

**Table 3 animals-12-01207-t003:** Mean score of agreement with statements on coexistence with hooded crows by the membership of hunting company of the respondents. Only 18 statements are shown for which the membership of hunting company difference was significant.

Statement No. (See Table 2)	Agreement Score				*F*	*p*	*η* ^2^
	Members (Mean)	Members (S.D.)	Non-members (Mean)	Non-members (S.D.)			
**2**	2.00	1.240	2.72	1.321	7.836	0.005	0.004
**3**	2.22	1.340	3.41	1.379	19.781	<0.001	0.011
**4**	2.19	1.302	3.13	1.260	14.976	<0.001	0.009
**5**	2.00	1.209	3.27	1.210	29.501	<0.001	0.017
**6**	3.07	1.639	3.98	1.138	16.639	<0.001	0.009
**7**	4.07	1.299	2.52	1.245	41.295	<0.001	0.023
**8**	1.96	1.160	3.35	1.210	35.017	<0.001	0.020
**9**	2.19	1.442	3.75	1.278	39.684	<0.001	0.022
**10**	1.67	1.109	3.19	1.252	39.725	<0.001	0.022
**11**	2.19	1.618	2.96	1.397	8.214	0.004	0.005
**13**	1.59	1.185	3.45	1.301	54.350	<0.001	0.030
**14**	4.30	0.912	3.08	1.219	26.783	<0.001	0.015
**15**	1.52	0.935	2.84	1.252	29.806	<0.001	0.017
**16**	1.74	1.318	2.97	1.279	24.623	<0.001	0.014
**18**	4.30	1.235	3.06	1.219	27.190	<0.001	0.015
**19**	2.78	1.368	1.93	1.139	14.501	<0.001	0.008
**20**	1.44	0.847	2.19	1.082	12.694	<0.001	0.007
**21**	3.85	1.292	3.09	1.353	8.441	0.004	0.005

Note: *F* denotes the value of *F* statistics, *p* denotes the computed significance value, while *η*^2^ denotes the value of effect size.

**Table 4 animals-12-01207-t004:** Percentages, mean, median and S.D. of acceptability of population management methods (listed in decreasing order of the sum of “Acceptable” and “Completely acceptable” values).

Management Method	Completely Unacceptable	Unacceptable	Neutral	Acceptable	Completely Acceptable	Mean	Median	S.D.	Efficiency/ Permissibility
Scaring crows using techniques that do not cause noise	12.6	9.5	27.4	25.7	24.8	3.41	4	1.297	Conditionally permitted
All measures for scaring crows	21.1	18.9	32	15	13	2.80	3	1.290	Conditionally permitted
Scaring crows with noise	28.8	18	26.1	15.9	11.2	2.63	3	1.342	Conditionally permitted
All measures to control the number of adult crows by authorized persons	25.2	18.7	29.3	14.2	12.6	2.70	3	1.325	Permitted
All measures to control nesting success	22.4	16.4	34.9	14.1	12.2	2.77	3	1.281	Not permitted
Use of traps to capture crows in urban areas	40.5	18.2	19.5	10.8	11	2.34	2	1.381	Conditionally permitted
Use of traps on farmland or in the countryside	42.8	18	20.6	9.4	9.2	2.24	2	1.334	Permitted from July 1 to February 28
Persecution of breeding crows using domesticated birds of prey (falconry)	47.2	14.9	21	9.9	7	2.15	2	1.300	Conditionally permitted
Physical removal of crows’ nests in urban areas	50.2	17.2	18.1	7.9	6.6	2.03	1	1.260	Conditionally permitted
Chemical sterilisation of crows	52.3	15.6	17.8	8.3	6	2.00	1	1.255	Permitted
Shooting adult crows on farmland or in the countryside	44.9	19.8	21.3	7.2	6.8	2.11	2	1.247	Permitted from July 1 to February 28
Removal of chicks or eggs from nests	68.7	13.7	10.1	3.9	3.6	1.60	1	1.051	Not permitted
Shooting adult crows in urban areas	69.7	13.3	10.8	2.8	3.4	1.57	1	1.019	Not permitted
Crows should be considered as rats and could be eliminated by anyone	68.8	14	11.9	2.8	2.5	1.56	1	0.974	Not permitted
Shooting crow chicks in nests	87.1	5	5.2	1.2	1.5	1.25	1	0.738	Not permitted
Setting poisoned bait for crows	82.2	9	6.2	1.2	1.4	1.31	1	0.759	Not permitted

**Table 5 animals-12-01207-t005:** The mean values of the clusters determined by the two-step clustering analysis.

Component	Mean Values of the Clusters		*F*	*p*	$\eta^{2}$
	Cluster 1 (47.5% of the Respondents)	Cluster 2 (52.5% of the Respondents)			
**(Dis)Interest in crows**	0.28	−0.25	130.57	<0.001	0.071
**Population management of crows**	0.75	−0.68	1773.93	<0.001	0.509
**Openess to different crow projects**	0.09	−0.08	11.36	0.001	0.007
**Hard persecution**	−0.39	0.35	271.96	<0.001	0.137
**Soft persecution**	−0.63	0.57	979.20	<0.001	0.364

Note: Clusters are determined by an automatic cluster number search algorithm based on a log-likelihood distance measure. The following variables were used for clustering: (dis)interest in crows, population management of crows, openness to different crow projects, hard persecution, and soft persecution components. The value of the Bayesian information criterion in the case of one, two, and three cluster solutions was the following: 6008.77, 5101.11, 4666.34.

**Table 6 animals-12-01207-t006:** The results of the contingency table analysis between clusters and species knowledge.

	Cluster 1	Cluster 2	Total
**Recognized all species (%)**	52.0 (3.1)	48.0 (−3.1)	100.0
**Not all species are recognized (%)**	44.3 (−3.1)	55.7 (3.1)	100.0

Note: Adjusted residuals are in parentheses.

## Data Availability

The data are freely available from the authors upon request.

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
