# Peer review of "Is the Hitchcock Story Really True? Public Opinion on Hooded Crows in Cities as Input to Management"

_animals, 2022, doi:10.3390/ani12091207_

Round 1
Reviewer 1 Report
The manuscript is well written and very interesting and should get published after a slight revision. Interactions between crows and people can be serious and urban people might take notice of the experience of nature by watching the nice crows. The hooded crow is a great, funny and very intelligent bird species. I miss in that study the positive effects of crows, e.g. some ecosystem services, like promoting environmental education by studying animal behaviour, eliminating carcass. Unfortunately, the questionnaire did not include those positive effects.
I do not know for what reason the PCA was calculated. I would drop it as well as table 6 and 8, as this information is not helpful.
The title is a bit lurid, but good. However, the name of the movie should be added and explained what happened in that movie. At the end, an answer should be given regarding the question raised in the title.
Minor comments
Line 98 – 115 Positive points are missing and should be added.
Table 1 could be added by a column displaying the average values for Hungary. This would help to better interpret the data. Hunting association not “hinting”.
Line 226-228. The information on Excel 2016 is not needed. Even you might use Word 2016 but did not mention.
Line 268-283: Could get shortened as it is just a repletion of the table. Two sentences explaining the most important points might be enough.
Table 3 and 4 could be merged.
Repetitions of the results should be omitted, e.g. line 387-390.
Chapter 4.2 should be shortened by half. It is not so interesting what happened in Slovenia.
Line 449: no brackets for 62.1%
Line 31: Which kind of problems are meant? Some examples are needed.
Line 54: The problems should be named here.
Line 629: The year is missing.
References were not formatted in the same way.
Reviewer 2 Report
Review of the paper „Is the Hitchcock story really true? The social judgment of the Hooded Crow”
The article presents the chances of minimizing conflicts between humans and crows in a very interesting way, based on the surveyed attitudes of people expressed in contact with these birds. Questionnaires in the field of human relationships with birds were filled in by users of social networks, which is an example of modern social science. Nevertheless, I believe that the main message of the manuscript should be better communicated. In the current version, I am not sure what the surveyed public opinion in the field of human conflicts with birds in urban areas is for. I found this information only in the last very short paragraph of the manuscript, while it could be placed earlier in the text.
The title itself is problematic, suggesting that the authors surveyed the social judgment of crows or that crows were judged by humans, and at the same time in the manuscript, readers learn that public opinion is rarely taken seriously. I believe that the article could be published, but it requires some corrections, including the organization of surveys’ description and earlier declaring how the public opinion can be used. I also suggest that you change the title and drop the phrase “social judgment” from the text to rather a social opinion or the real impact of crows on society. I propose a title: “Is the Hitchcock story really true? The potential of real public opinion to protect conflict birds.”
The article is illustrated by numerous tables which are a less friendly tool for communication with readers in comparison with plots. Below are some other suggestions that could improve the proposed article. However, I would like to emphasize that the overall impression of the manuscript is good and my comments result from a positive attitude to the presented issue.
Simple Summary. In general, I think that the confidence that the society is able to decide on the most optimal bird management strategy in urban areas is a far-reaching conclusion from the authors. I suggest changing the way that society plays a role in managing bird populations – public consultation can be valuable, but the decision should be made by naturalists and urban area managers who have data on bird population status and the scale of the problem in urban areas. Sometimes people have a negative view of harmless birds. I suggest highlighting the fact that the voice of society should go to professionals and be verified by them starting from the Simple Summary.
Abstract. L31 – please, be specific and add some examples of these problems. L32 – please add information on who decides how to manage bird populations in studied urban areas (for example, I live far from Hungary and do not have this knowledge). L36-37 – please, be specific – define what “not negative” and “not serious conflicts” mean. L42 – the same as sterilization of crows?
Introduction. L63 – what „wide range of problems” – please, be specific. L104-105 – please, rephrase („not always happy”…). L116-132 – this information is very valuable and interestingly described; please use some fragments already in the earlier stages of the manuscript (as I recommended above). L140-141 – please, rephrase. As mentioned, "social judgment" does not sound well in scientific manuscripts concerning nature conservation.
Materials and Methods. There should be 2.1. Please, add some information on whether those social media groups were related to the problems of a given city, or were they created to collect general information, e.g. historical or memories from the local community? If these are groups concerned with various (environmental) problems in a given area, there is a possibility of less objective opinions because of specific users of this kind on groups. If these L172-174 – was the quiz to check if the person answering the questions recognizes the hooded crow? L191-193 – fair point, do the authors have any data or observations that conflicts between people and animals can be caused by people who do not usually use the Internet, e.g. the poor people, old hunters? Table 1 – I would write the demographic and resident data of the respondents. The table caption does not explain what the sign near the word “women” means.
Research combining social science and nature conservation has become more frequent in recent years and is becoming a separate field of science. I propose to emphasize this fact in the proposed manuscript because it proves the originality of the conducted research.
Statistical analyses sound reasonable.
Results. Data from Tables 2 and 3 can be illustrated as bar graphs that would be more reader-friendly. In the case of Table 3, the charts can show e.g. 3-4 most interesting results from the questionnaires. Tables 6 and 8 should be organized as two PCA plots, in my opinion.
Discussion. L387 – urban areas? This chapter is the best part of the manuscript, although I suggest adding short separate „Conclusions”.
I also suggest that authors use the past tense in the manuscript, or at least think about it in the chapters "Results" and "Discussion".
